# Sex-specific DNA methylation associations with circulating urate levels and BCG-induced urate changes
Zhaoli Liu [1,2], Tania O. Crişan[3,4], Cancan Qi[1,2], Manoj Kumar Gupta[1,2], Xuan Liu[1,2], Simone J. C. F. M. Moorlag[3], Valerie A. C. M. Koeken[1,2,3,5], Xun Jiang [1,2], Mohamad Ballan[1,2], L. Charlotte J. de Bree[3], Vera P. Mourits[3], Xu Gao[6,7], Andrea Baccarelli[6,8], Joel Schwartz [9], Frank Pessler[1,10], Carlos A. Guzmán[1,11], Yang Li [1,2,3,12,13], Mihai G. Netea [3,14], Leo A. B. Joosten [3,4,15] & Cheng-Jian Xu [1,2,15] ✉

## Abstract

**Background** Urate concentration and the physiological regulation of urate homeostasis exhibit clear sex differences. DNA methylation has been shown to explain a substantial proportion of serum urate variance, mediate the genetic effect on urate concentration, and co-regulate with cardiometabolic traits. However, whether urate concentration is associated with DNA methylation in a sex-dependent manner is unknown. Additionally, it is worth investigating if urate changes after perturbations, such as vaccination, are associated with DNA methylation in a sex-specific manner.

**Methods** We investigated the association between DNA methylation and serum urate concentrations in a Dutch cohort of 325 healthy individuals. Urate concentration and DNA methylation were measured before and after *Bacillus Calmette-Guérin* (BCG) vaccination, used as a perturbation associated with increased gout flares. The association analysis included united, interaction, and sex-stratified analysis.

**Results** 215 CpG sites are associated with serum urate in males, while 5 CpG sites are associated with serum urate in females, indicating sex-specific associations. Circulating urate concentrations significantly increase after BCG vaccination, and baseline DNA methylation is associated with differences in urate concentration before and after vaccination in a sex-specific manner. The CpG sites associated with urate concentration in males are enriched in neuro-protection pathways, whereas in females, the urate change-associated CpG sites are related to lipid and glucose metabolism.

**Conclusions** Our study enhances the understanding of how epigenetic factors contribute to regulating serum urate levels in a sex-specific manner. These insights highlight the importance of personalized and sex-specific approaches in medicine.

## Plain language summary

Men and women have different levels of urate, a metabolic byproduct that can crystallize and cause gout. Small chemical changes to our DNA, known as DNA methylation, play a role in biological functions and can change over time, unlike fixed genetic factors. In this study, we aim to explore the link between blood urate levels and DNA methylation by recruiting 325 healthy individuals before and after they received a BCG vaccine. We find that urate is linked to DNA methylation differently in men and women, and that urate levels rise after vaccination in a sex-specific way. These findings suggest that the body regulates urate differently in men and women. Understanding these differences could help guide more personalized treatments for urate-related disorders.

Urate is the end product of purine metabolism in humans[1,2] and has been shown to have beneficial effects, including acting as an antioxidant[3], enhancing the immune response[4], and reducing the risk of neurodegenerative disease[5]. However, an inappropriately elevated concentration of urate (adult male > 0.42 mmol/L, adult female > 0.36 mmol/L), known as hyperuricemia, can lead to the development of gout and other health conditions[6–8]. Therefore, elucidating the

regulatory mechanisms governing circulating urate concentration is crucial.

Previous research has identified both host and environmental factors that can influence circulating urate concentration[8,9]. While a genome-wide association study (GWAS) has identified 183 loci associated with urate concentration, these loci only explained about 7.7% of the variance in serum urate concentration[10], suggesting that other non-genetic factors may play an

important role. DNA methylation is one of the well-studied epigenetic markers[11], and modifications to DNA methylation record an individual's response to environmental exposures[12]. A recent large-scale trans-ancestry epigenome-wide association study (EWAS) identified 99 urate-associated Cytosine-phosphate-Guanine (CpG) sites, which collectively account for 11.6% of serum urate variance[12]. This underscores the crucial role of epigenetic regulation in urate concentration. While sex differences in circulating urate concentrations and their association with genetic, cardiometabolic traits, and sex hormones are well studied[13–15], the extent to which the epigenome is differentially associated with urate concentrations in males and females is not well understood.

Vaccines have been widely used to prevent infections, yet recent reports suggest that vaccination might increase the risk of gout flares[16,17]. Gout is the main disease which is primarily associated with hyperuricemia[18,19]. However, whether vaccination impacts circulating urate concentrations or the exact mechanism by which they could drive gout flares remains unknown. In this report, we focus on the *Bacillus Calmette-Guérin* (BCG) vaccine, commonly used as an anti-tuberculosis vaccine, to investigate the epigenetic mechanisms of interactions between vaccination and urate metabolism. Prior studies have indicated an association between anti-tuberculosis drugs and hyperuricemia[20,21], prompting the exploration of whether the BCG vaccine, as an anti-tuberculosis vaccine, is similarly associated with increased urate levels. BCG vaccination may potentially promote a proinflammatory environment, thereby contributing to the development of inflammation-mediated diseases. Moreover, it induces trained immunity, which could modulate the immune response in gout and inflammasome activation triggered by urate and urate crystals[18,22].

To investigate the interplay among epigenetic signatures, circulating urate concentrations, and vaccination, we recruited 325 Dutch individuals from the Human Functional Genomics Project (300BCG cohort). Genome-wide DNA methylation profiles and urate concentrations were measured for all participants before BCG vaccination, as well as two weeks and three months after vaccination. We conducted the sex-specific epigenome-wide association study (EWAS) of urate concentration and vaccination-induced urate changes, respectively, including CpG sites from both autosomes and the X chromosome. Utilizing baseline data before vaccination, we identify a sex-specific association between serum urate and DNA methylation. Specifically, in males, urate-associated CpG sites are predominantly enriched in genes linked to pathways such as neuroprotection. After BCG vaccination, we observe a significant increase in serum urate concentration at 2 weeks, which remains elevated for up to three months. Moreover, we observe a sex-specific epigenetic association with BCG-induced urate changes. In females, urate change-associated CpG sites are enriched in genes related to lipid and glucose metabolism. These identified CpG sites are further validated in three additional cohorts. Our study provides valuable insights into how epigenetic associated with urate concentration before and after vaccination, which may enable us to better understand sex-specific epigenome co-regulation of urate and other urate-related diseases.

## Methods
### Cohort information
The 300BCG cohort ($n = 325$, 56% females) is a healthy western descent population-based cohort from the Human Functional Genomics Project, with an average age of 26 years (±10 standard deviation). Volunteers were recruited between April 2018 and June 2018 in the Radboud University Medical Center as described previously[23,24]. All volunteers received a standard dose of 0.1 mL BCG (Intervax, Canada, strain BCG-Bulgaria) intradermally in the left upper arm by a medical doctor. Whole blood was drawn before BCG vaccination (baseline, day 0), 2 weeks (day 14), and 3 months (day 90) after vaccination. Blood was collected in the morning. Exclusion criteria include: acute or chronic illness at the time of sampling; a medical history of immunodeficiency; any febrile illness within 4 weeks before participation; previous BCG vaccination or having lived in tuberculosis endemic countries; history of tuberculosis; any vaccination 3 months before participation; use of systemic medication other than oral contraceptives or

acetaminophen; use of antibiotics 3 months before inclusion. This study was approved by the Ethical Committee of Radboud University Medical Center (NL58553.091.16). All participants gave written informed consent.

This study had three replication cohorts, all European ancestry, including 500FG, NAS, and BCG booster. 500FG cohort is a part of the Human Functional Genomics Project. Within 500FG, 260 participants (58% females) who had both baseline DNA methylation value and circulating urate concentration available were included in this study, with a mean age of 27 years (±12 standard deviation). NAS (Normative Aging Study, dbGaP Study Accession: phs000853.v1.p1) consists of 774 males (0% females) with a mean age of 72.65 years (±6.82 standard deviation). The summary statistic which recorded the EWAS of urate was obtained for the replication analysis. The NAS cohort, which consisted solely of males, was used to replicate the urate-associated CpG sites identified in this male population. The detailed information of the BCG booster cohort was described previously[25]. Within this cohort, 13 females (100% females) who received single-dose BCG at day0 were included in this study, with an average age of 26 years (±5 standard deviation). DNA methylation and serum urate concentrations were measured before and 90 days post BCG vaccination. The BCG booster cohort was used to replicate the urate change-associated CpG sites identified in females.

To explore if urate increase is BCG vaccination unique, we included TIV cohort. The TIV cohort is described in detail previously[26]. The main study was conducted between September 2015 to May 2016. 200 individuals (age: 65–80 years) were randomly selected from the residents' registration office in Hannover and received an adjuvanted trivalent inactivated influenza vaccine (TIV).

### DNA methylation quantification and quality control
DNA purification from whole blood was done by QIAamp DNA blood kits (Qiagen Benelux BV, Venlo, the Netherlands). The DNA concentration was measured using a NanoDrop spectrophotometer at 260 nm. High-quality DNA was used for genome-wide DNA methylation profile by either Infinium© MethylationEPIC array (BCG and BCG booster) (~850,000 CpG sites) or MethylationEPIC v2 array (500FG) (~937,055 CpG sites). The DNA methylation values were gained from the raw IDAT files using the minfi package in R (v.4.2.0)[27]. We excluded poor quality and sex-mismatched samples. For the probes on the autosomes, stratified quantile normalization[28] was performed after filtering out bad quality probes with a detection $P$ value > 0.01, cross-reactive probes, polymorphic probes[29], and probes on the sex chromosomes. Problematic probes due to mapping inaccuracies and flagged probes in the EPIC v2 array provided by Illumina were also removed in the 500FG cohort. For the probes on the X chromosome, functional normalization was applied[30]. After all the quality control steps above, in the BCG cohort, 858 samples (286 individuals for three-time points; females = 160, males = 126), 751,564 probes on the autosome, and 16,724 probes on chromosome X remained for further analyses. In the BCG booster cohort, 26 samples (13 females from two time points) and 794545 probes on autosome remained for further analysis. In the 500FG cohort, we got 296 samples and 854,447 probes on autosomes for further analysis.

### Circulating urate concentration measurement
BCG and BCG booster cohort: Serum was separated from whole blood by centrifuging at 1500 $g$, 30 min. The serum urate concentration was measured by Radboud Laboratory Diagnostic (RLD), with a Roche C8000 system using the module C702. The serum urate value from 935 samples (the number of samples from each time point: baseline: 321; day 14: 314; day 90: 300) was successfully obtained for further analysis.

500FG and TIV cohort: Circulating urate concentrations were obtained from an untargeted metabolism dataset which was described previously[31,32]. To confirm if we could use the urate concentration from the untargeted metabolism dataset in the 500FG and TIV cohorts, we did the following analysis: in our 300BCG cohort, we have urate concentrations at baseline from both Radboud Laboratory Diagnostics (RLD) and untargeted

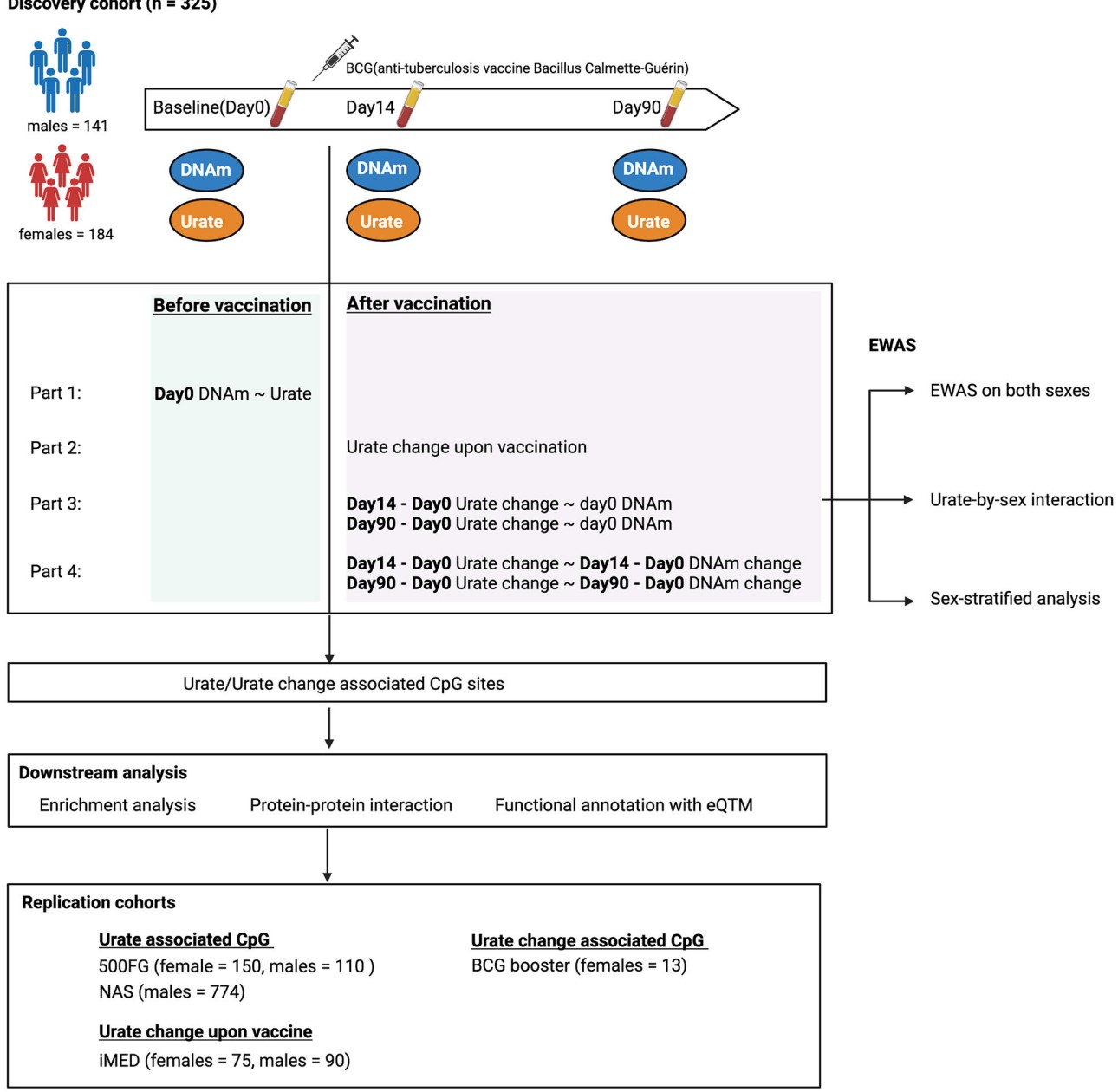

**Fig. 1 | The workflow of this EWAS on urate concentrations.** In this study, we included one discovery cohort called 300BCG, DNA methylation (DNAm) and urate data were generated from this cohort. We did the following analyses in our study. First, we explored whether the baseline serum urate was associated with baseline DNA methylation in a sex-specific way by interaction and sex-stratified analysis. Second, we investigated the urate change and performed the EWAS of urate change after BCG. We did the EWAS on both sexes, urate-by-sex interaction EWAS, and sex-stratified EWAS. After obtaining the urate or urate change-associated CpG sites, we annotated these sites with their proximal genes, followed by enrichment analysis and protein-protein interaction analysis. We also linked these CpG sites with gene expression using published eQTM results. Independent cohorts were used to validate the identified CpG sites.

metabolism. The values from these two methods were highly correlated (Supplementary Fig. 1).

**Epigenome-wide association analyses (EWAS)**
Before performing EWAS, we excluded the methylation outliers (value < 25 quantile − 3*IQR or value > 75 quantile + 3*IQR). We performed the EWAS as the previous study did[33]: (1) EWAS of baseline urate: To explore how serum urate concentration is associated with methylation, baseline urate concentration was analyzed as the independent variable with M value as the dependent variable in a robust linear regression model adjusting for age, sex, cell type proportions, and batch effect. (M value ~ urate + age + sex + batch + cell type); (2) Sex interaction EWAS: To investigate the sex effect on the association between urate and methylation, we added sex*urate as an interactive factor in our robust linear regression model (M value ~ urate + age + sex + batch + cell type + sex*urate). The P value representing whether 'sex * urate' from the linear model had a significant impact on methylation was recorded; (3) EWAS of urate change after BCG vaccination: To assess the association between baseline methylation and urate change after vaccination, a robust linear model was performed for each CpG site, with rank normalized urate change as the dependent variable, baseline methylation value as the independent variable, age, sex, batch, and cell proportion as the covariates (urate change ~ M value (Day 0) + age + sex + batch + cell type); (4) Sex stratified EWAS: We separated all participants into females and males. The baseline urate-EWAS and urate change-EWAS

were conducted in each sex group; The model was M value ~ urate + age + batch + cell type and urate change ~ M value + age + batch + cell type. (5) Association between urate change and DNAm change: DNA methylation change was the difference in methylation residuals which was calculated by regressing out cell proportion and batch at each time point detailed in our previous publication[34], followed by fitting into the model as: urate change ~ DNAm change + age + sex. Significant CpG sites were identified as those with an adjusted p-value below 0.05 after multiple testing correction.

## Downstream analysis

IlluminaHumanMethylationEPICanno.ilm10b4.hg19 R package and Genomic Regions Enrichment of Annotations Tool[35] were used to annotate the identified CpG with their proximal genes. Enrichment analyses were performed using the online tool ConsensusPathDB (CPDB, http://cpdb.molgen.mpg.de)[36]. Protein-protein interaction networks functional enrichment analysis was done using STRING.

## The association analysis between hormone concentrations and urate change after BCG vaccination

Baseline circulating testosterone, 17-hydroxy progesterone, androstenedione, cortisol, 11-deoxy cortisol were measured by liquid chromatography-tandem mass spectrometry as described previously[37,38]. The Spearman method was used to estimate the correlation between hormones and urate change.

## Metabolites from the metabolism pathway in TIV cohort

Metabolome profiling was done using flow-injection mass spectrometry as mentioned previously[32]. In detail, Plasma samples were distributed randomly across assay plates to minimize bias related to vaccine response, time point of collection, and sex. For metabolite extraction, 20 μl of each plasma sample was mixed with 180 μl of 80% methanol in a deep-well extraction plate, a process performed by General Metabolics (Boston, USA). The mixture was vortexed for 15 s, incubated at 4 °C for 1 h, and then centrifuged at 3750 rpm for 30 min at 4 °C. Metabolomic profiling was conducted using flow-injection mass spectrometry. Each sample was injected twice in rapid succession (within 0.96 min) to generate technical replicates. To ensure consistency and monitor instrument performance, a pooled reference sample was periodically injected during the batch. Sample acquisition within each plate was randomized.

## Statistics and reproducibility

EWAS were conducted using robust linear regression models to account for potential outliers. All CpG sites were analyzed individually, and p-values were corrected for multiple comparisons using the Benjamini-Hochberg method. CpG sites with a false discovery rate (FDR)-adjusted p-value < 0.05 were considered statistically significant. The sample size for each analysis is reported in the corresponding results or figure legends.

## Reporting summary

Further information on research design is available in the Nature Portfolio Reporting Summary linked to this article.

## Results

### Cohort information and study design

Our discovery cohort (300BCG[23,24,39,40]) involves 325 Dutch individuals (56% female). Measurements are taken at three-time points: baseline, 14 days after, and 90 days after BCG vaccination. We assess whole blood genome-wide DNA methylation (DNAm) of 850,000 CpG sites and serum urate concentrations (Fig. 1, Supplementary Fig. 2, Supplementary Data 1). To replicate significant associations between urate concentrations and DNA methylation, we include three additional cohorts (500FG[41,42], n = 260; NAS[43], n = 774; BCG booster[25], n = 13) as replication cohorts, in which both methylome, urate levels and BCG induced urate changes are measured. Details on these cohorts are provided in the Methods section. The analysis workflow is presented in Fig. 1.

## Discovery and replication of sex-specific urate-associated CpGs at baseline

Univariate associations between principal components (PCs 1-20) derived from the DNA methylation data and covariates, including estimated cell proportion of six cell types by Houseman's method[44] are shown in Supplementary Fig. 3. The top 20 PCs collectively accounte for 30.34%, 30.17%, and 33.70% of the variance in DNAm at baseline, 14 days, and 90 days post BCG vaccination, respectively. Cell proportions, batch (sample plate), and age are significantly associated with the top 5 PCs, whereas smoking status, BMI, height, and weight do not show any significant correlation (Supplementary Fig. 3). Next, we assess the association of urate concentration with DNA methylation in all participants in the 300BCG cohort at baseline (day 0) (Fig. 2a). Although no genome-wide significant CpG site is identified (False Discovery Rate (FDR) adjusted P value < 0.05) (Fig. 2b (1)), 81 out of 96 urate-associated CpG sites reported previously[12] exhibit a consistent direction of the estimated effect (Supplementary Fig. 4a).

Given the significant difference in baseline urate concentrations between males and females observed (P value < 2.2e−16, Wilcoxon rank sum test, Fig. 2c), we examine the influence of sex on the DNA methylation-urate association. We employ two approaches: urate-by-sex interaction analysis and sex-stratified analyses. The interaction analysis identifies six CpG sites exhibiting significant interaction effects (Fig. 2b (2), Supplementary Fig. 4b). The CpG site with the strongest interaction effect is cg03227576 (P value = 1.95e−08), mapping to the PAPPA gene, which encodes a protein involved in inflammation[45], systemic glucose homeostasis[46], and female fertility[47]. Intriguingly, sex-stratified analyses reveal 215 genome-wide significant CpG sites in males compared to only 5 CpG sites in females (Fig. 2b (3-4), Supplementary Data 2, 3). This finding may suggest a potential sex-specific epigenetic association of urate concentrations. The comparable inflation factors for both models (1.04 in females vs 1.11 in males) indicate that this difference is unlikely to be due to model inflation. Furthermore, Fig. 2d highlights the distinct effects between males and females. Gene enrichment analysis of the 215 urate-associated CpG sites annotated genes in males reveals enrichment in pathways related to cell projection, neuron part, learning and memory, system development, and response to stimulus terms (Fig. 2e). Consistent with this, the protein-protein interaction network analysis based on genes annotated to significant CpG sites in males shows functional enrichments in the neuron projection and axon (cellular component in Gene Ontology) (Fig. 2f). When we check the association between identified CpG sites and gene expression using the Biobank-based integrative omics study (BIOS) database[48] (Supplementary Data 4), we find that cg05231308 significantly correlates with UCN gene expression, which is implicated in both neurodegenerative conditions and skeletal system disorders. Additionally, cg09930046 correlated with the expression of AZU1, a crucial player in innate immunity, functioning as both a neutrophil granule-derived antibacterial factor and a monocyte- and fibroblast-specific chemotactic glycoprotein[49]. Conversely, protein-protein interaction analysis in females shows enrichment in cell cycle-associated pathways (Supplementary Fig. 4c).

Urate-associated CpG sites identified from males are enriched in IRF1, Foxa2, Atf1, Fli1, IRF4, and Hoxd10 motifs. Urate-associated CpG sites identified from females are enriched in MYNN, Bapx1, Nkx2.5, ETV4, and Nkx3.1 motifs (Supplementary Fig. 4d, e). Many of which are involved into the immune responses, metabolism, and cell differentiation. For example, for the motifs IRF1 and IRF4, these are involved in the regulation of immune response and may influence inflammatory pathways, which are known to be associated with urate metabolism and gout. For the motifs Foxa2 and Foxa3, these transcription factors are crucial for liver function and metabolic processes. The enrichment near these CpG sites might indicate a role in hepatic regulation of urate levels. ETV4 involves in the regulation of cell proliferation, differentiation, and survival.

We seek to validate the urate-associated CpG sites identified in males from the 300BCG cohort using two independent cohorts: the 500FG (n = 260) and the NAS cohort (n = 774). In the 500FG, 72 out of 130 CpG sites show the same direction of association with urate concentrations, with

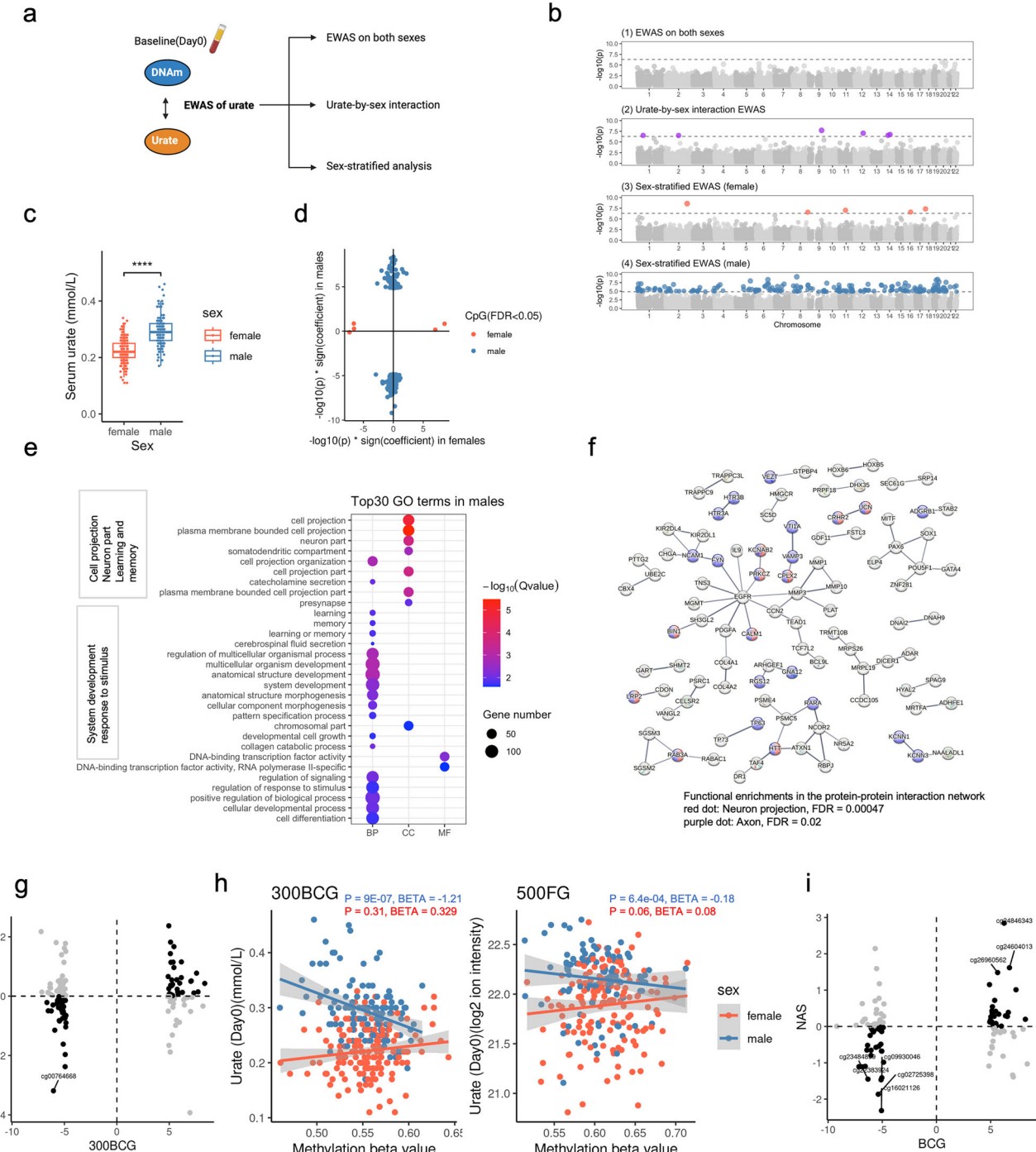

**Fig. 2 | Serum urate concentrations were associated with DNA methylation in a sex-specific manner. a** Schematic representation of the analysis flow. **b** The Manhattan plot shows the results from EWAS of urate in all participants (1), urate-by-sex interaction analyses (2), females (3), and males (4) respectively. The CpG sites were ordered by their chromosomal position on the x-axis with their -log10(P value) of the association on the y-axis. The dotted horizontal lines represented the level of significance corrected for multiple testing. colorful dot: significant CpG sites with FDR < 0.05, grey dot: non-significant CpG sites. $n = 281$(females = 158, males = 123) independent samples were included in this analysis. **c** Bar plot showing the difference of the serum urate level at baseline between males and females. $n = 321$(females=184, males=137) independent samples were included in this analysis. **d** The scatter plot showing the distribution of -log10(P value) * sign(effect estimate) of the significant sites (FDR < 0.05) from urate-EWAS in males and females. The color represented the sites from different sex groups. **e** Dot

plot describing the GO enrichment categories of genes annotated to the significant CpG sites identified in males. The color and size of the dot indicate the significance and the number of annotated genes in the given GO term, respectively (BP: biological process, MF: molecular function, CC: cellular component). The top 30 enriched categories were shown. **f** Protein-protein interaction networks functional enrichment analysis of the urate-associated CpG sites from males. **g** Scatter plot showing the distribution of -log10(P value) * sign (effect size) of the male urate-associated CpG sites in BCG and 500FG cohorts. Black dots represented CpG sites with the same effect direction, while grey dots represented CpG sites with different effect directions. **h** One example of the replicated CpG site. **i** Scatter plot showing the distribution of -log10 (P value) * sign (effect size) of the male urate-associated CpG sites in BCG and NAS cohorts. Black dots represented CpG sites with the same effect direction, while grey dots represented CpG sites with different effect directions.

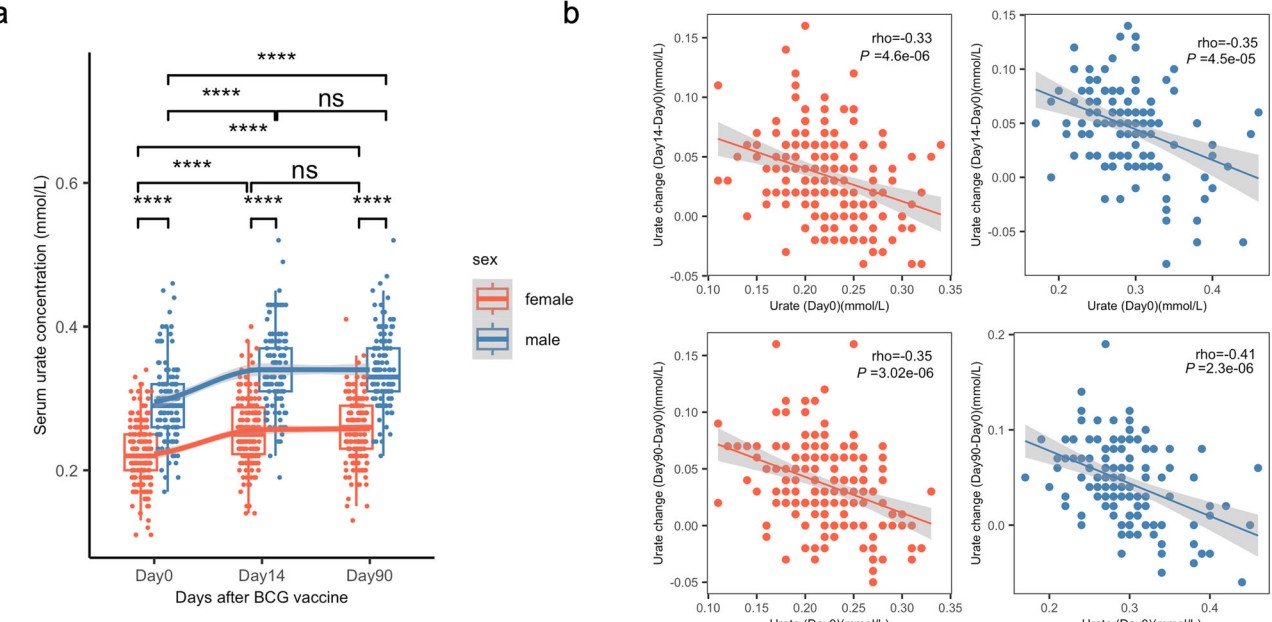

**Fig. 3 | Serum urate significantly increased after BCG vaccination and ΔUrate was negatively correlated with baseline urate. a** Boxplot showing the serum urate concentration at different time points, color indicating different sex groups (n = 281 independent individuals, 3 time points paired). The paired samples Wilcoxon test was used to compare the urate concentration among different time points. Wilcoxon rank sum test was used to compare the difference between males and females. ns: not significant, *P < 0.05, **P < 0.01, ***P < 0.005, ****P < 0.001. **b** Spearman's correlation between ΔUrate and Urate_baseline. The color indicated different sex groups.

eight sites reaching nominal significance (P value < 0.05). One CpG cg00764668 achieves an FDR significance (FDR = 0.042) and is located on chr19, mapping to the *KCNN1* gene (Fig. 2g). *KCNN1* is associated with intellectual developmental disorder, potassium channels, and transmission across chemical synapses[50]. Notably, the methylation level at cg00764668 exhibits a negative correlation with circulating urate concentration in males but not in females (Fig. 2h). In the NAS, 57 out of 107 CpG sites show the same direction, with eight sites reaching nominal significance and no CpG site reaching FDR significance (Fig. 2i). Next, when we validate the findings of urate-DNA methylation in males by meta-analysis with 500FG and NAS, 12 out of 63 CpG sites show the same direction across all these three cohorts, including the cg00764668 mentioned above. Two CpG sites, cg22383924 and cg24846343, reach nominal significance (Supplementary Data 5). cg22383924 is annotated to gene *TP73*, which plays a role in cellular responses to stress and development[51]. cg24846343 maps to gene *DDTL*, which may have lyase activity[52]. Replication of CpG sites identified in females from the 300BCG cohort is performed using the 500FG cohort. Two out of four CpG sites display the same direction of association, with one site (cg00864618) reaching nominal significance. cg00864618 maps to gene *CHRAC1*, which plays a role in DNA transcription, replication, and packaging with other histone-fold proteins[53].

In summary, a sex-specific association between circulating urate concentrations and DNA methylation is observed. Males show a strong methylation-urate signal, with 215 significant CpG sites versus only 5 in females. These male-specific CpG sites are enriched in pathways related to neuronal function and immune regulation.

**Discovery and replication of sex-specific CpGs associated with BCG-induced urate change**

As shown in Fig. 3a, BCG vaccination induces a significant increase in urate concentrations (median value in females: Urate_baseline = 0.22 mmol/L, Urate_day14 = 0.25 mmol/L, P value < 2.2e-16; median value in males: Urate_baseline = 0.29 mmol/L, Urate_day14 = 0.34 mmol/L, P value < 2.2e-16. Paired samples Wilcoxon test). Notably, these elevated urate concentrations persist for up to three months after vaccination (median value in females: Urate_day90 = 0.26 mmol/L; median value in males: Urate_day90 = 0.33 mmol/L)

(Fig. 3a). Interestingly, the magnitude of the urate rise differs by sex, with males exhibiting a higher increase in both short-term (P value = 0.00058, One-sided Wilcoxon rank sum test) and long-term (P value = 0.01, One-sided Wilcoxon rank sum test) than female (Supplementary Fig. 5a). Both short-term and long-term urate change show a significantly negative correlation with Urate_baseline in both sexes (Fig. 3b).

Next, we assess the association between baseline DNA methylation and BCG-induced urate change in the 300BCG cohort (Fig. 4a). We first conduct an EWAS between baseline DNAm and urate change at 14 days post-vaccination (short-term change) for all participants. This analysis does not identify any epigenome-wide association CpG sites (Fig. 4b (**1**)). As urate change after BCG vaccination differs between males and females, we stratify the participants by sex and identify sex-specific CpG sites associated with urate change: 63 in females and 67 in males (Fig. 4b (**2-3**), Supplementary Data 6, 7). Notably, the effect size patterns of these CpG sites are distinct between males and females (Fig. 4c), suggesting sex-based differences in epigenetic associations. The CpG site with the strongest association in females is cg21182196 (P value = 5.46e−09), mapping to the *CLMN* gene. This gene might be involved in neuron projection development and its expression could be regulated by vitamin D3[54]. In males, the most significant CpG site is cg03088047 (P value = 1.06e−09), mapping to *TBC1D22A*, a gene involved in regulating lipid homeostasis[55].

In females, urate change-associated CpG sites are linked to genes involved in vitamin D metabolism, lipid and glucose metabolism, and SLC-mediated transmembrane transport processes (Fig. 4d, Supplementary Data 8). Among the 34 CpG sites with a negative association with urate increase, 13 are mapped to the genes related to metabolism, such as genes in the lipid metabolism pathways (*PCYT1A*; *ALDH3B2*; *DHCR7*; *SPNS2*; *GC*; *PGS1* and *SETD4*). Notably, 17 CpG sites are annotated to genes in the SLC-mediated transmembrane transport process and peroxisome pathway. On the other hand, the 29 CpG sites with a positive association with urate increase are annotated to the genes involved in the insulin signaling pathway (*PRKAG2*; *RHEB*; *YWHAQ* and *SOCS1*). Interestingly, pathways related to the translocation of SLC2A4 (GLUT4) to the plasma membrane are enriched, suggesting these sites are linked to glucose metabolism. Additionally, the enrichment of synaptic signaling pathways associated with autism

### EWAS of BCG induced urate change in discovery cohort

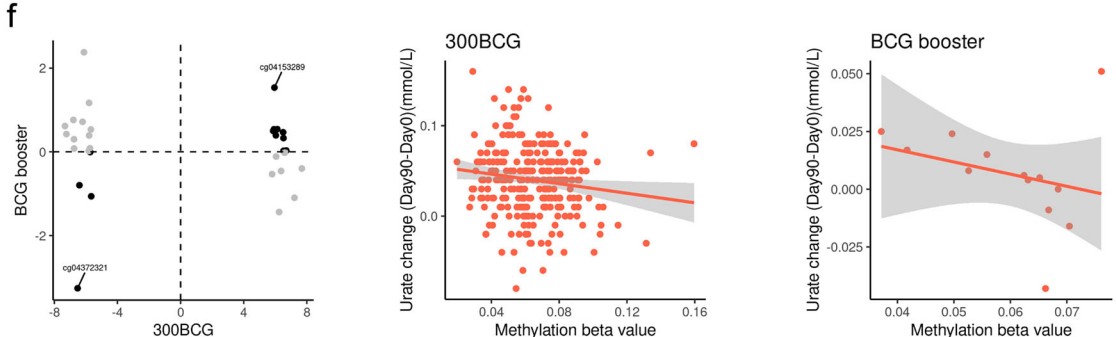

### Replicated urate change-CpG site

**Fig. 4 | ΔUrate associated with DNAm$_{baseline}$ in a sex-specific manner and urate change-associated CpG sites were related to human metabolism. a** Schematic representation of the analyses. **b** The Manhattan plot showing the results from EWAS of Δurate$_{short}$ in all participants (1), females (2), and males (3) respectively. The CpG sites were ordered by their chromosomal position on the x-axis with their -log10 (P value) of the association on the y-axis. Colorful dot: significant CpG sites, grey dot: non-significant CpG sites. **c** The scatter plot showing the distribution of -log10 (P value) * sign(effect estimate) of the significant sites from EWAS of Δurate$_{short}$ in males and females. Color represented the significant sites from different sex groups. Bar plot describing the KEGG enrichment categories of genes annotated to the significant CpG sites identified from the EWAS of Δurate$_{short}$ in females (**d**) or males (**e**). "BETA < 0" represented the significant sites with a negative effect estimate. "BETA > 0" represented the significant sites with a positive effect estimate. **f** One example of the replicated urate change-associated CpG site. The y-axis represented the urate change from day90 minus day0 and after rank normalization. The x-axis represented the beta value of cg04372321.

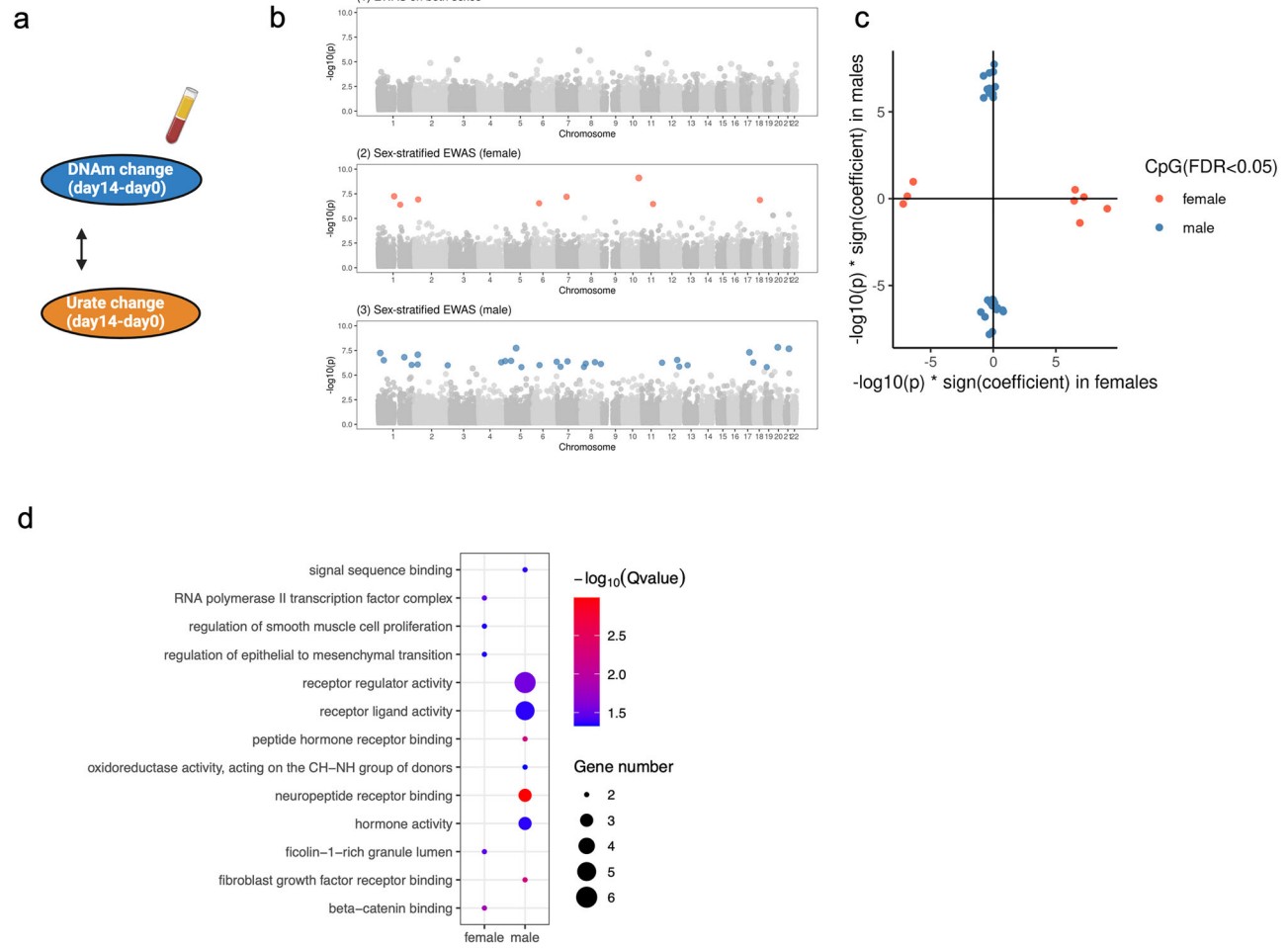

**Fig. 5 | ΔUrate was associated with ΔDNAm upon BCG vaccination in a sex-specific manner. a** Schematic representation of the association analyses. **b** The Manhattan plot shows the results from the epigenome-wide association between $\Delta\text{Urate}_{short}$ and $\Delta\text{DNAm}_{short}$ in all participants (1), females (2), and males (3) respectively. The CpG sites were ordered by their chromosomal position on the x-axis with their -log10 (P value) of the association on the y-axis. Colorful dot: significant CpG sites, grey dot: non-significant CpG sites. **c** The scatter plot showing the distribution of -log10 (P value) * sign(effect estimate) of the significant sites from the epigenome-wide association between $\Delta\text{Urate}_{short}$ and $\Delta\text{DNAm}_{short}$ in males and females. The color represented the significant sites from different sex groups. **d** Dot plot describing the KEGG enrichment categories of genes annotated to the significant CpG sites identified from the epigenome-wide association between $\Delta\text{Urate}_{short}$ and $\Delta\text{DNAm}_{short}$. The color and size of the dot indicated the significance and the number of annotated genes in each category.

spectrum disorder points to potential neuromodulatory roles (Fig. 4d, Supplementary Data 8). In males, the genes annotated to those CpG sites with a positive correlation with urate increase are enriched in the cardiac progenitor differentiation pathway. Conversely, CpG sites with a negative correlation with urate increase are linked to various metabolic pathways, including amino acid metabolism (Methionine De Novo and Salvage Pathway), fatty acid metabolism (e.g., Linolenic acid metabolism), and other lipid metabolism pathways related to acyl chain remodeling (Fig. 4e, Supplementary Data 9).

A sex-specific epigenomic association with urate change is also observed three months post-BCG vaccination (long-term change). Moreover, it is notable that there is a high consistency in the effect direction, with males showing 95.5% and females showing 100% consistency between EWAS of short-term urate change and long-term urate change. We identify 33 CpG sites associated with long-term urate change after BCG in females and 112 CpG sites in males (Supplementary Fig. 5b, Supplementary Data 10, 11), and observe the distinct pattern of effect size between males and females (Supplementary Fig. 5c). In males, cg13148076 shows the strongest association with long-term urate change. This probe maps to *PATZ1*, which plays an important role in regulating T cell development[56]. In females, the probe with the greatest association, cg09755770, maps to gene *RCSD1*, which is involved in the cellular hyperosmotic response[57]. We validate these

identified long-term urate change-associated CpG sites using an independent cohort (300BCG booster cohort, *n* = 13 females), Thirteen out of 33 CpG sites exhibit the same direction as the discovery cohort, and one site (cg04372321, mapped to *DGKQ*) reached to FDR significance (FDR = 0.01) (Fig. 4f). *DGKQ* mediates the regeneration of phosphatidylinositol from diacylglycerol in the PI-cycle, further suggesting that the urate change-associated CpG sites are linked to human lipid metabolism.

We next investigate the association between urate change (ΔUrate) and DNAm change (ΔDNAm) following BCG vaccination. We first examine the association between short-term urate change $\Delta\text{Urate}_{short}$ and short-term DNA methylation change $\Delta\text{DNAm}_{short}$ (Fig. 5a). No epigenome-wide association is identified from the EWAS conducted on all participants. However, sex-stratified analyses reveal 29 CpG sites associated with $\Delta\text{Urate}_{short}$ in males and 8 in females (FDR < 0.05) (Fig. 5b, Supplementary Data 12, 13). Notably, the strongest association in males is observed for cg00471000 (P value = 1.52e−08), mapping to *BPIFB6* on chromosome 20, a key regulator of secretory pathway trafficking and viral replication[58]. In females, the strongest association is seen for cg04660100 (P value = 7.80e−10), mapping to *HABP2* on chromosome 10. The protein encoded by this gene is involved in cell adhesion. Figure 5c depicts the distinctive $\Delta\text{DNAm}_{short}$-$\Delta\text{Urate}_{short}$ association profiles between males and females. Functional enrichment analysis of the significant CpG sites in males reveals

enrichment for pathways related to receptor regulator activity, receptor-ligand activity, and hormone activity (Fig. 5d). We then assess the association between long-term urate change ($\Delta \text{Urate}_{long}$) and long-term DNA methylation change ($\Delta \text{DNAm}_{long}$). Two CpG sites, cg09451092 mapped to *CCDC19* (encoded-protein enables AMP binding activity) and cg04372321 mapped to *DGKQ*, are identified from the EWAS on all individuals. Similar to the short-term analysis, sex-stratified analysis reveals 52 CpG sites associated with $\Delta \text{Urate}_{long}$ in males and 6 in females (FDR < 0.05) (Supplementary Data 14, 15). Males and females exhibit distinct patterns of association (Supplementary Fig. 6a, b). Functional enrichment analysis of the significant CpG sites from females indicates a link to the small molecule biosynthetic process (Supplementary Fig. 6c).

Collectively, circulating urate levels increase upon BCG vaccination, with greater and more sustained effects observed in males. Importantly, DNA methylation at baseline and post-vaccination are associated with both short- and long-term changes in urate levels in a sex-specific manner. Sex-stratified analyses reveal numerous CpG sites linked to urate dynamics: up to 112 CpG sites in males and 33 in females for long-term changes. These CpG sites map to genes involved in metabolic regulation, immune function, neuronal signaling, and transmembrane transport, highlighting sex-dependent epigenetic mechanisms that may underlie individual variability in metabolic responses to vaccination.

### Circulating hormones may explain the sex-differential effect, but not CpG sites on chromosome X

To investigate the factors contributing to the observed sex-related differences in urate change after BCG vaccination, we first examine baseline characteristics such as age and body mass index (BMI). However, these factors do not differ significantly between males and females (Supplementary Data 1). Next, we hypothesize that the sex-differential effect may be explained by CpG sites located on sex chromosomes, which are not modeled in our EWAS on autosomes. Therefore, we re-normalize the methylation data by adding data from the X chromosome. Subsequently, we perform association analyses between urate (or urate change) and CpG sites located on the X chromosome. Following the association analysis, we identify seven CpG sites associated with urate change in females and three in males on the X chromosome, which achieve an X-chromosomal wide FDR significance (Supplementary Data 16). However, when comparing the ratio between X chromosome and autosomes, we do not find significant enrichment on the X chromosome compared to autosomes ($P = 0.6$ in females, $P = 1$ in males, Fisher's exact test).

We next investigate whether circulating hormone concentrations might contribute to the sex-specific effects. We analyze the correlation between baseline hormone concentrations (androstenedione, cortisol, 11-deoxy cortisol, 17-hydroxy progesterone, and testosterone) and baseline urate concentrations or urate change post-vaccination in both males and females. As expected, most of these circulating hormones, except cortisol, display sex-based differences in baseline concentrations (Supplementary Fig. 7a). Notably, none of these circulating hormones correlate with baseline urate concentrations in either males or females (Supplementary Fig. 7b). Intriguingly, baseline plasma cortisol, 11-deoxy cortisol, and 17-hydroxy progesterone show a negative association with long-term urate change ($\Delta \text{Urate}_{long}$) after BCG vaccination only in females (cortisol: $P$ value = 0.0006, 11-deoxy cortisol: $P$ value = 0.0016, 17-hydroxy progesterone: $P$ value = 0.0152, Spearman's correlation), but not in males (Fig. 6a). Furthermore, this association is not seen with short-term urate change ($\Delta \text{Urate}_{short}$) after BCG vaccination (Supplementary Fig. 7c).

### Urate increase upon vaccination is not specific to BCG

To determine whether vaccine-induced urate increase is unique to BCG, we analyze data from an independent cohort of 165 elderly individuals (mean age: 72 years ± 4 standard deviation), who receive an adjuvanted trivalent inactivated influenza vaccine (TIV, from iMED cohort). Plasma metabolome is profiled before vaccination (baseline, day 0), 7 days, and 21 days post-vaccination. Notably, the influenza vaccine significantly increases

circulating urate concentration in females, but not males (Fig. 6b), suggesting a sex-specific response and a potential mechanism beyond BCG-specific effects. It is also worth noting that both short-term and long-term urate change show a significantly negative correlation with $\text{Urate}_{baseline}$ in both sexes (Supplementary Fig. 8). Further investigation of the TIV cohort reveals significant changes in the majority of metabolites within the purine metabolism pathway after vaccination (Fig. 6c). These findings suggest that a shift in purine metabolism may contribute to the increased urate levels following vaccination.

## Discussion

In this study, we investigate the sex-specific relationships between DNA methylation and circulating urate concentrations and the effect of BCG vaccination on circulating urate concentrations. We observe significant sex differences in the epigenetic regulation of urate concentrations. Specifically, in males, baseline urate-associated CpG sites are predominantly enriched in genes linked to neuroprotection. Furthermore, we observe an increase in serum urate concentrations post-BCG vaccination in both sexes. Urate change-associated CpG sites in females are linked to lipid and glucose metabolism.

Sex differences in circulating urate concentrations and their association with genetic and cardiometabolic traits are well known. The important role of epigenetic regulation on circulating urate concentration is supported by numerous EWAS findings, such as the recent study from Tin et al.[12]. However, until now, results from sex-stratified EWAS of urate have not been reported, but are clearly present in our study. For example, one reported CpG site (cg11266682), which is located on gene *SLC2A9* and has a causal effect on urate, presents a significant association with urate only in females ($P = 0.0058$) but not males ($P = 0.62$) in our discovery cohort. This finding is consistent with the GWAS results that the genetic variants within this locus are associated with lower serum urate concentrations among women only[15]. The urate-associated CpG sites from Tin et al.[12] are linked to learning and neurological function, whereas in this study, we observe that this pathway is specifically observed in males at baseline before vaccination. However, after an intervention like BCG vaccination, we find that urate change-associated CpG sites are linked to neuromodulatory pathways in females. Although the hypothesis that elevated urate is associated with a lower risk of neurodegenerative disease only in males has been proposed in the literature[59,60], evidence in the literature remains mixed, with some studies suggesting that urate exhibits neuroprotective effects in both sexes[61,62]. Our study provides evidence of a sex-specific DNA methylation signature of serum urate and neurological function in whole blood before and after vaccination. However, the results of this study do not provide direct evidence supporting a causal relationship between serum urate and neuroprotection.

Our study further unveils the sex-specific epigenetic association between BCG-induced urate change and baseline methylation. The SLC-mediated transmembrane transportation pathway is negatively associated with urate increase exclusively in females. Among the CpG annotated genes enriched in this pathway, *SLC24A3* is a sodium/calcium exchanger. *SLC45A1* and *SLC4A4* play a role in glucose uptake and glucose homeostasis. Additionally, genes related to insulin signaling like *PRKAG2; RHEB; YWHAQ* are positively associated with an increase in urate circulating concentration. These findings suggest the epigenetic co-regulation of urate increases after BCG, and glucose metabolism is more prominent in females. The association between higher urate concentrations and the incidence of glucose tolerance, type 2 diabetes development, hypertension development, and lipid accumulation patterns differs between the sexes[63,64]. Our findings provide insight that epigenetic co-regulation might be a general mechanism underlying the observed pleiotropy between urate and cardiometabolic traits.

Several pieces of evidence suggest that sex hormones may play a role in sex-dimorphisms of urate. Female hormones are thought to play a role in

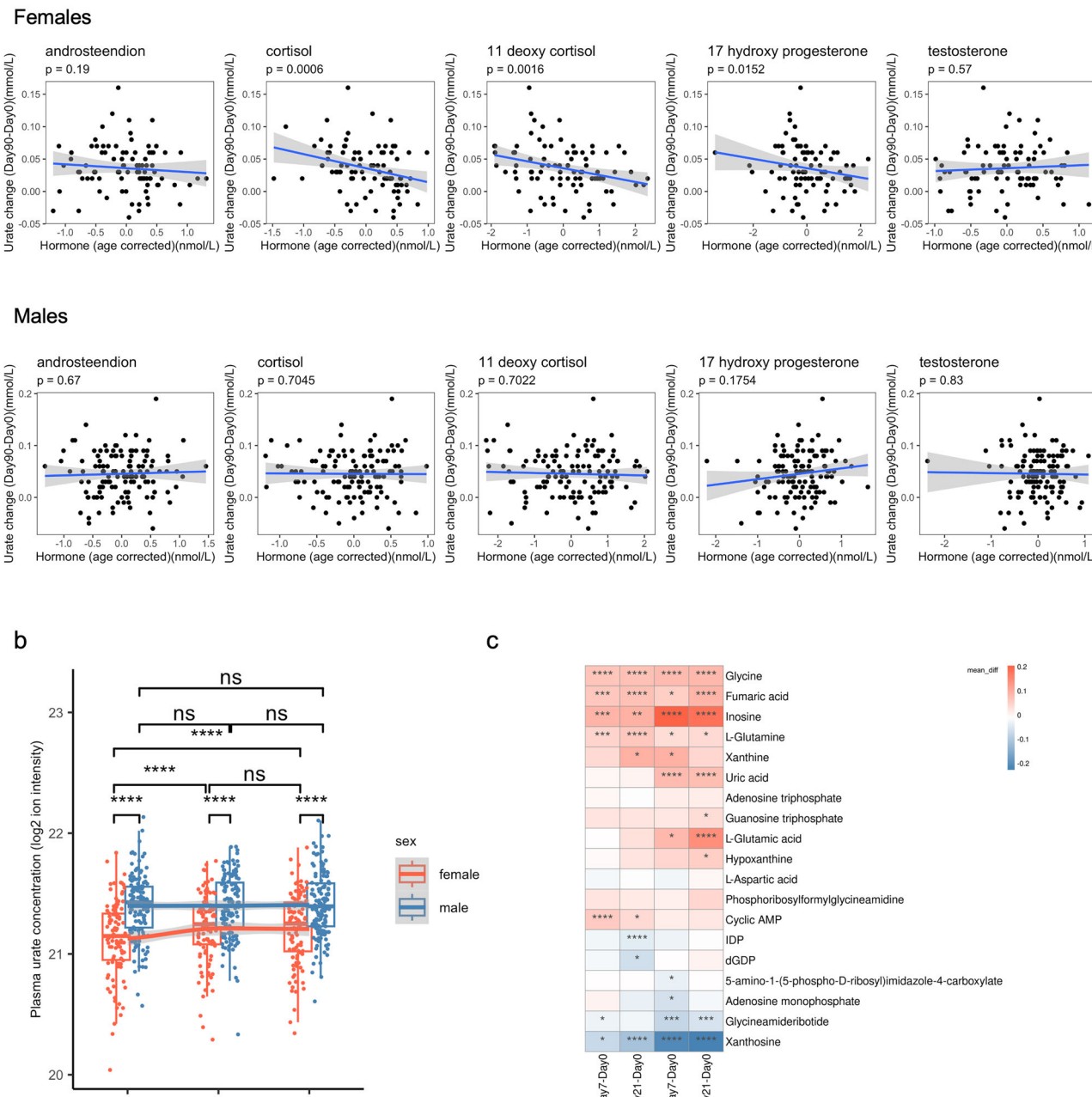

**Fig. 6 | The increase in urate levels following vaccination is not unique to BCG, and it may be influenced by shifts in purine metabolism. a** Spearman's correlation between $\Delta\text{Urate}_{\text{long}}$ and baseline hormone level (after age correction). **b** Boxplot showing the plasma urate concentration at different time points, color indicating different sex groups. The paired samples Wilcoxon test was used to compare the urate level among different time points ($n = 165$ independent individuals, 3 time points paired). Wilcoxon rank sum test was used to compare the difference between males and females. ns: not significant, *$P < 0.05$, **$P < 0.01$, ***$P < 0.005$, ****: $P < 0.001$. **c** The mean difference of metabolites level between each two time points ($n = 165$ independent individuals, 3 time points paired). These metabolites are all that we can find from the purine metabolism pathway. Stars represented the significance of the difference using paired samples Wilcoxon test. *: $P < 0.05$, **$P < 0.01$, ***$P < 0.005$, ****$P < 0.001$.

urate elimination and regulating renal urate transporters[65]. In this study, we show that steroid hormones, including cortisol, 11-deoxy cortisol, and 17-hydroxy progesterone, are associated with the changes in urate concentrations after BCG vaccination (day 90 vs. day 0) in females exclusively. One previous study presents that steroids are necessary to maintain a normal glomerular filtration rate (GFR) and renal plasma flow (RPF)[66]. Our results suggest that hormones might engage in controlling and regulating long-term urate increase in females.

The purine metabolite pathway is a potential contributor to the increase in urate concentrations upon BCG vaccination. Previous studies show that the co-accumulated metabolites module, associated with BCG-induced trained immunity responses, is enriched in the purine

**Article**

metabolism[23]. The interplay between epigenetics and metabolites may play an important role in the induction, regulation, and maintenance of trained immunity[67]. Furthermore, increased urate leads to immune reprogramming and inflammasome activation, which contribute to the development of inflammation-mediated diseases such as atherosclerosis, reactive arthritis, as well as other autoimmune and autoinflammatory disorders[18,68,69]. Our investigation into the effects of influenza vaccination on circulating urate concentration reveals that this urate-elevating phenomenon is not specific to BCG, but rather a more universal feature associated with vaccination and immune stimulation. This is further supported by recent studies reporting that some routinely administered vaccines, including the COVID-19 vaccine and recombinant zoster vaccine, are associated with increased gout flares[16,17,70], thereby pinpointing the possibility of urate concentration increases after the administration of these vaccines. Notably, urate levels increase in both sexes after BCG vaccination in our discovery cohort, which primarily consists of young adults. However, in our TIV cohort of elderly individuals, urate levels specifically increase in females following influenza vaccination, highlighting the role of aging in urate homeostasis.

Our findings hold potential translational implications in several important areas. First, by revealing sex-specific DNA methylation association of serum urate levels before and after vaccination, our study provides a foundational step toward the development of precision medicine approaches that consider sex as a biological variable. Given the known link between elevated serum urate and increased risk of cardiometabolic and neuroinflammatory diseases, our results may help in identifying individuals—particularly males—with epigenetic profiles that predispose them to urate-driven pathologies. Second, the observed urate elevation following BCG and influenza vaccination suggests that changes in urate metabolism may serve as a biomarker of immune activation or metabolic stress post-vaccination. This could have implications for monitoring vaccine responses or side effects in vulnerable populations, such as the elderly or those with metabolic syndrome. Lastly, our study highlights hormone-epigenome interactions in urate regulation, particularly in females, pointing to a possible role for hormone-modulating therapies in managing urate-related disorders. Further longitudinal and interventional studies will be needed to validate these findings and translate them into clinical applications.

This study has some limitations. First, the sample size is relatively small, and future studies should involve larger cohorts for validation. Additionally, the NAS and TIV cohorts have different age ranges compared to the discovery cohort, and since urate levels change with age, this may have limited the replication performance. Second, although we study over 800,000 CpG sites of the human methylome, this still only captures about 3% of all human CpG sites[71]. Third, our research mainly focuses on a few steroid hormones. Further research should explore the role of estrogens and serum parathyroid hormone (PTH) in urate regulation after vaccination. Forth, in the 500FG and BCG booster cohorts, urate levels are derived from metabolite datasets rather than direct serum measurements. While these two methods show a high degree of correlation, we believe that maintaining consistent serum urate measurements for validation will enhance replication performance. Fifth, the epigenome-wide association between whole blood DNA methylation and serum urate concentrations may not represent urate homeostasis in other important organs such as the kidney or liver. Multi-tissue studies might help us to capture a more comprehensive picture of the urate epigenetic regulation. Finally, while our study focuses on DNA methylation in whole blood, this captures one important aspect of the epigenetic landscape. Other regulatory mechanisms, such as histone modifications and chromatin accessibility, are not assessed in this analysis. Future studies incorporating complementary epigenomic approaches, such as ATAC-seq or ChIP-seq, will be important to provide a more comprehensive view of the molecular pathways regulating urate homeostasis and its modulation in response to vaccination.

In conclusion, this study reveals sex-specific epigenetic mechanisms underlying urate regulation. These findings improve our understanding of urate biology and may inform the development of sex-specific therapies targeting urate for improved health outcomes.

## Data availability

The methylation data are available in the European Genome-phenome Archive (EGA) under accession number EGAS00001007498. All other source data (including serum urate measurements, phenotypic information, and analysis datasets used to generate figures) are available at figshare under the link: https://figshare.com/articles/dataset/source_data_for_publication_Sex-specific_DNA_methylation_associations_with_circulating_urate_levels_and_BCG-induced_urate_changes/29412995.

## Code availability

The code can be accessed via https://doi.org/10.5281/zenodo.15734441[72].

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

## Acknowledgements
CJX is supported by the Lower Saxony MWK Sprung Fund (19777006) and Deutsche Forschungsgemeinschaft (DFG) Fund (497673685). LAB and TOC were supported by a grant from the Romanian Ministry of European Investments and Projects (P_37_762, MySMIS 103587). YL is supported by an ERC Starting Grant (948207). MGN is supported by an ERC Advanced Grant (833247) and a Spinoza Grant of the Netherlands Organization for Scientific Research. ZL was supported by a grant from the China Scholarship Council. Regarding 300BCG and 500FG cohorts, we would like to thank all volunteers for their participation in the study. We thank all colleagues who were involved in the sample collection and data generation. Regarding the TIV cohort, we would like to thank the study teams at the Clinical Research Center Hannover, in particular, Ms. Katrin Paul (study nurse) and at the Hannover Unified Biobank for their contribution to data collection and processing of biosamples. We thank Ms. Sabrina Wieghold (Study Center Hannover of the German National Cohort health study), and Ms. Conni Senske (Helmholtz Centre for Infection Research, Braunschweig, Germany) for supporting the recruitment process, and Nina Burgdorf, Mohamed A. Tantawy, and Aaqib Sohail (TWINCORE, Hannover, Germany) for technical support. Regarding the NAS cohort, the molecular analyses in the US Department of Veterans Affairs (VA) Normative Aging Study have been supported by the U.S. National Institute of Environmental Health Sciences (NIEHS) (R01ES015172, R01ES021733). The VA Normative Aging Study is supported by the Cooperative Studies Program/ERIC, US Department of Veterans Affairs, and is a research component of the Massachusetts Veterans Epidemiology Research and Information Center (MAVERIC). Additional support to the VA Normative Aging Study was provided by the US Department of Agriculture, Agricultural Research Service (contract 53-K06-510). The views expressed in this paper are those of the authors and do not necessarily represent the views of the US Department of Veterans Affairs.

## Author contributions
C.J.X., L.A.B., Y.L. and M.G.N. conceptualized and designed the study. Z.L. performed the data analysis supervised by C.J.X., LA.B., Y.L. and M.G.N. Q.C., M.K.G., X.L., X.J. and M.B. helped with the validation of the findings. T.O.C., L.A.B., M.G.N., S.M., V.K., C.B. and V.M. helped with the recruitment of participants and biological data collection from the BCG cohort. X.G., A.B., and J.S. contributed to the sample recruitment and data analyses from the NAS cohort. F.P. and C.A.G. coordinated the TIV cohort and obtained the corresponding samples. C.J.X. and Z.L. wrote the manuscript with input from all the authors. All authors reviewed and approved the manuscript.

## Funding

## Competing interests
The authors declare no competing interests.

## Additional information

¹Centre for Individualised Infection Medicine (CiiM), a joint venture between the Helmholtz-Centre for Infection Research (HZI) and Hannover Medical School (MHH), Hannover, Germany. ²TWINCORE, a joint venture between the Helmholtz-Centre for Infection Research (HZI) and Hannover Medical School (MHH), Hannover, Germany. ³Department of Internal Medicine and Radboud Center for Infectious Diseases (RCI), Radboud University Medical Center, Nijmegen, the Netherlands. ⁴Department of Medical Genetics, Iuliu Hațieganu University of Medicine and Pharmacy, Cluj-Napoca, Romania. ⁵Research Centre Innovations in Care, Rotterdam University of Applied Sciences, Rotterdam, the Netherlands. ⁶Department of Environmental Health, Mailman School of Public Health, Columbia University, New York, NY, USA. ⁷Department of Occupational and Environmental Health Sciences, School of Public Health, Peking University, Beijing, China. ⁸Harvard T.H. Chan School of Public Health, Boston, MA, USA. ⁹Department of Environmental Health, Harvard T.H. Chan School of Public Health, Boston, MA, USA. ¹⁰Research Group Biomarkers for Infectious Diseases,

TWINCORE, a joint venture between the Helmholtz-Centre for Infection Research (HZI) and Hannover Medical School (MHH), Hannover, Germany. [11]Department Vaccinology and Applied Microbiology, Helmholtz-Centre for Infection Research (HZI), Braunschweig, Germany. [12]Cluster of Excellence RESIST (EXC 2155), Hanover Medical School, Hannover, Germany. [13]Lower Saxony center for artificial intelligence and causal methods in medicine (CAIMed), Hannover, Germany. [14]Department for Immunology and Metabolism, Life and Medical Sciences Institute (LIMES). University of Bonn, Bonn, Germany. [15]These authors contributed equally: Leo A. B. Joosten, Cheng-Jian Xu. ✉e-mail: Xu.Chengjian@mh-hannover.de

