## [Transparent Peer Review file · Communications Medicine]

Sex-specific DNA methylation associations with circulating urate levels and BCG-induced urate changes

Corresponding Author: Professor Cheng-Jian Xu

This manuscript has been previously reviewed at another journal from Communication series. The manuscript was considered suitable for publication without further review at Communications Medicine.

Version 0:

Reviewer comments:

Reviewer #1

(Remarks to the Author)

Liu and Colleagues have assessed the association between circulating urate concentrations and DNA methylation before and after BCG vaccination, to identify sex specific differences and vaccination induced changes in associations between CpG sites and urate levels. This work is well executed, the manuscript well written and the findings are novel and interesting. I have only two comments:

- 1) the title is a bit ambiguous; first, 'epigenetic signatures of circulating urate' reads as if urate itself has epigenetic signatures. Second, it is not 100% clear what 'its' refers to. Please improve the title.
- 2) The results section is quite dense with a large number of different methylation sites and gene association being listed. For clarity and ease of reading the authors are encouraged to add some summarizing/concluding sentences at the end of each results section with the key take home messages.

Reviewer #2

(Remarks to the Author)

This is a well-designed paper where authors have shown how epigenetic factors contribute to regulate serum urate levels in a sex-specific manner. However, I am concerned about the translational relevance of the study. The authors should discuss that in more detail and include a paragraph in discussion.

The authors have detected whole blood DNA methylation which will not be a representation of true homeostasis in tissues. Epigenetics is a wide term which covers histone modifications as well. If the authors prefer to use the word "Epigenetics" they should also investigate histone modifications associated by either ATAC-seq or ChIP-seq.

Version 1:

Reviewer comments:

Reviewer #1

(Remarks to the Author)

The authors have sufficiently addressed my concerns and I have no further comments

Reviewer #2

(Remarks to the Author)

The authors have addressed all my comments. I recommend this manuscript for publication.
